# Beyond Classification: Elaborating Network Predictions for Better Weakly Supervised Quantization

**Chih-Chieh Chen**[1]                                                    JACKFRANK@GMAIL.COM
**Chang-Fu Kuo**[1,2,3]                                                   ZANDIS@GMAIL.COM

[1] *Center for Artificial Intelligence in Medicine, Chang Gung Memorial Hospital, Taoyuan, Taiwan*

[2] *Medical Education Department, Chang Gung Memorial Hospital, Taoyuan, Taiwan*

[3] *Division of Rheumatology, Allergy and Immunology, Chang Gung Memorial Hospital, Taoyuan, Taiwan*

**Editors:** Under Review for MIDL 2026

## Abstract

For clinical applications, more detailed information such as specific locations and the region of interest (ROI) volumes is preferred. However, most of the time only classification annotations are available. Class Activation Mapping (CAM) and its variants are the most commonly used techniques for weakly supervised localization tasks. In this study, we assessed both traditional and modern network architectures regarding classification accuracy and CAM visualization. Although all networks achieved high AUROC scores and their heatmaps closely corresponded to pathology locations, we observed that the heatmaps were influenced by the particular network architectures and pretrained weights used. Additionally, current models produce heatmaps from small latent spaces (e.g. $16 \times 16$), which limits the precision of these heatmaps for further detailed analysis.

Based on the observations mentioned above, we designed a UNet-style architecture that utilizes pretrained classification networks as the encoder and produces heatmaps within a latent space of size $128 \times 128$. We observed that the generated heatmaps are more detailed and suitable for weakly supervised segmentation. We validated the effectiveness of our approach using the intracerebral hemorrhage (ICH) dataset.

**Keywords:** ICH classification, heatmap generations, weakly supervised localization.

## 1. Introduction

The localization of pathologies is just as crucial as their classification for clinical use. By utilizing heatmaps or bounding boxes, clinicians can revisit patient data to verify the presence of pathologies in specific areas, thereby lowering the false negative rates in diagnoses. However, creating bounding box annotations is time-consuming. In this work we are interested in ICH subtype segmentation. Hematoma volume and hematoma expansion are essential predictors of mortality and outcome (Li et al., 2024). Previous studies (Scherer et al., 2016) have shown that the ABC/2 method, which estimates lesion volume by multiplying the largest total clot diameter by the applicable perpendicular clot diameter and the number of slices, frequently yields inaccurate overestimations of ICH volume. This indicates a clear need for the development of a more precise volume estimation method.

Consequently, it is valuable to explore how weakly supervised methods can minimize the need for extensive labeling. In medical image analysis, Grad-CAM based methods (Zhou et al., 2016; Selvaraju et al., 2017) are probably currently the most widely used techniques.

For Grad-CAM, the heatmap of the chosen layer by weighted summing the channel values, where the weight of the channel is given as the summations of partial derivatives of final outputs with respect to all the spatial positions. Due to the spatial biases inherent in convolutional neural networks, locations corresponding to class-related features tend to have higher output values and growth rates, resulting in elevated heatmap scores at these positions.

In this study, we assessed the Grad-CAM approach on quantifying subtypes of intracranial hemorrhage (ICH). The five subtypes considered are intraparenchymal hemorrhage (IPH), intraventricular hemorrhage (IVH), subarachnoid hemorrhage (SAH), subdural hematoma (SDH), and epidural hematoma (EDH). Typically, these subtypes can be differentiated by their distinct locations and shapes (Heit et al., 2016). We surprisingly found the outputs of fine-grained pretrained models are highly accurate and delicate. Therefore, our idea is to modify the heatmaps generated by Grad-CAM. However, since they are derived from low-resolution latent spaces, their representational capacity is limited.

Modern architectures achieve impressive discriminative performance with fewer parameters and FLOPs by incorporating modern components like the squeeze-and-excitation block (Hu et al., 2018) and depthwise convolutions, along with improved training and scaling methods. Building on this insight, our work aims to leverage features from hidden layers to produce more detailed heatmaps. To do this, we construct a U-shaped network that uses the pretrained model as the encoder. Since segmentation labels are unavailable, inspired by (Pinheiro and Collobert, 2015), we replace the final average pooling and linear layers with a Log-Sum-Exp (LSE) pooling layer, which sums the exponentials of patch outputs to generate predictions. The decoder's architecture is simple, and our intuition is to use the pretrained encoder, possibly with regularization terms, to guide the nearly linear, untrained decoder in producing more fine-grained heatmaps. We also incorporate intermediate supervision and regularization on each layer's output to ensure the generated heatmaps align with human intuition.

We demonstrated the effectiveness of our methods using ICH datasets. Our model was trained on the RSNA intracranial hemorrhage dataset (Flanders et al., 2020) and its performance was assessed on the BHSD dataset (Wu et al., 2023). Our empirical results indicate that although the classification accuracy of our proposed architecture is comparable to or surpasses that of standard models, the heatmaps produced by the Grad-CAM algorithm are significantly more accurate.

In sum, our contributions are as follows:

- We empirically highlight both the strengths and limitations of the Grad-CAM algorithm when applied to clinical tasks.

- To produce more detailed heatmaps, we introduced a method that also leverages the shallow layers of a pretrained network by constructing a UNet-like architecture and generating heatmaps from the final layer of this U-shaped network.

- We evaluated our proposed architecture on the classification of intracranial hemorrhage subtypes, showing that it achieves comparable classification performance while producing more precise heatmaps.

## 2. Related Works

### 2.1. ICH Segmentation

In recent years, deep learning-based approaches have been proposed. Authors in (Kuo et al., 2019) developed a joint classification and segmentation framework based on 4,396 CT scans and demonstrated a case-level sensitivity of 100% and specificity of 90% on 200 randomly selected CT scans. In their related work (Kuo et al., 2018), DICE score of 76.6% was reported. In a similar study (Monteiro et al., 2020), edema was included as an additional ground truth class. A simplified annotation was used, merging SDH, EDH, and SAH into a single category named extra-axial hemorrhage (EAH). The testing dataset, which excludes lesions of 1 mL or smaller, yielded a case-level mean Dice score of 59.3% for ICH.

Although these methods yield promising results, they require extensive fine-grained annotations. Therefore, it is desirable to create techniques that demand less annotation effort.

### 2.2. Grad-CAM

Given a classifier ending with a global average pooling layer followed by a fully connected layer, the final output corresponding to class $c$ is given by

$$y^c = w_l^c \cdot \frac{1}{Z} \Sigma_i \Sigma_j A_{i,j}^l, \tag{1}$$

where $A_{i,j}^l$ represents the output of the $l$-th feature at spatial location $(i, j)$ in the last convolutional layer and $w_l^c$ is the weight of the fully connected layer associated with class $c$ for unit $l$. In (Zhou et al., 2016), the heatmap $M^c$ is proposed to be

$$M_c(i,j) = \Sigma_l w_l^c A_{i,j}^l. \tag{2}$$

In (Selvaraju et al., 2017), the idea is further generalized to arbitrary neural networks by letting

$$w_l^c = \frac{1}{Z} \Sigma_i \Sigma_j \frac{\partial y^c}{\partial A_{i,j}^l}, \tag{3}$$

and the equivalence of these two approaches when the target layer is right before the final global classification layer is also demonstrated in (Selvaraju et al., 2017).

Lastly, to create visualizable heatmaps, $M_c$ in Equation 2 will be passed through a ReLU layer and further normalized by its maximum value.

To generate High resolution images, authors in(Selvaraju et al., 2017) further suggested combining Grad-CAM with guided backpropagation (Springenberg et al., 2014). For guided backpropagation, negative gradients are not backpropagted through ReLU layers when calculating the heatmaps with respect to image pixels. In (Selvaraju et al., 2017) it is suggested to use the Hadamard product of heatmap generated by Grad-CAM and by guided backpropagation to obtain a refined heatmap..

Integrated gradient (Sundararajan et al., 2017) is another common choice for generating high resolution heatmaps. Given an image $x$, reference image $x'$ (usually a black image), function $F$ and pixel $x_i$, in (Sundararajan et al., 2017), the heatmap score at $x_i$ is generated by integrating $\frac{\partial F(x' + (x - x'))}{\partial x_i}$ through $\alpha$ from 0 to 1.

Numerous studies have been conducted to enhance performance across various specialized domains (Smilkov et al., 2017; Djoumessi and Berens, 2025). In this work, we will choose guided backpropagation and intergrated gradient as our baseline model.

### 2.3. Weakly Supervised Segmentation

Weakly supervised learning is an encouraging approach in situations where annotated data is scarce, and numerous research methods have been explored (Chen and Sun, 2025). In the current era, with the advent of foundation models (Radford et al., 2021; Kirillov et al., 2023; Siméoni et al., 2025), it is possible to achieve more detailed weakly supervised segmentation outcomes. Nevertheless, even in the absence of foundation models, several significant studies have already shown highly promising results on datasets such as PASCAL VOC2012 (Everingham et al., 2010). For instance, by starting with Grad-CAM based predictions, in (Wang et al., 2018; Shen et al., 2018), performances are improved using techniques like the neural version of seed region growing (Adams and Bischof, 1994) or by grabcut (Rother et al., 2004). Although these methods might potentially yield good results in weakly supervised ICH segmentation tasks, we also wish to highlight that these modification techniques might assume a distinct boundary between class objects and the background. In contrast, this work aims to introduce a framework that does not rely on this assumption.

(Rasoulian et al., 2023) investigate weakly supervised ICH-segmentation approach using Swin transformer. A novel layer attention map with respect to the head of the self attention layer is proposed. However, to the best of the author's knowledge, the work works on ICH, namely, weakly supervised binary segmentation. Instead, in this work attempt to work on weakly supervised multi-label subtype segmentation. The motivations and purposes between (Rasoulian et al., 2023) and our work are also different. While pre-trained weights are not used in (Rasoulian et al., 2023), we aim to show that the outputs of contemporary network architectures, such as ResNet-RS and RDNet used in our study, with pre-trained weights, encode rich information and suitable for visualization purposes. Our primary contribution is to extract this information into higher resolution outputs, enabling us to obtain quantified data, such as subtype volume.

### 2.4. LSE Pooling Layer

Let

$$LSE_r(x_1, \ldots, x_n) = \log(\Sigma_i \exp(r \cdot x_i)) \tag{4}$$

be the scaled LogSumExp function. Recall that the partial derivatives

$$\frac{\partial LSE_r(x1, \ldots, x_n)}{\partial x_j} = \frac{\exp(r \cdot x_j)}{\Sigma_i \exp(r \cdot x_i)}. \tag{5}$$

Thus, when the global average pooling layer is replaced by the LSE pooling layer, according to Equation 3, the contributions to the weight $w_l^c$ with respect to the spatial locations are proportional to their scaled softmax values. This implies that the final heatmap scores at each spatial location are proportional to the product of their patch-level outputs and the scaled softmax values. Compared to heatmaps generated by networks using global average pooling, these heatmaps are significantly more sensitive to patch-level outputs.

## 3. Proposed Architecture and Algorithm

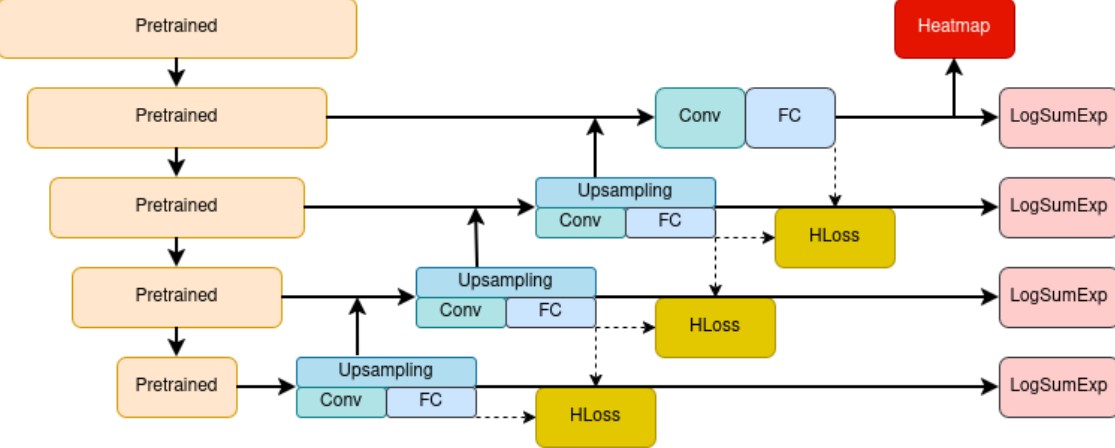

Figure 1: Proposed Architecture. HLoss refers to the heatmap consistency loss in Sec. 3.3

### 3.1. Observations on Heatmaps Generations

Our primary focus is on two groups of architectures: ResNet (He et al., 2016), DenseNet (Huang et al., 2017), along with their variants such as ResNet-RS (Bello et al., 2021) and RDNet (Kim et al., 2024). We trained these models to classify ICH subtypes, and all achieved AUROC scores exceeding 0.95. For ICH subtype localization, we observed that all models could produce heatmaps accurately pinpointing the locations of each subtype, especially when multiple subtype features are completely separate (see Fig. 2 for more details.)

Regarding the estimation of stroke volume subtypes, as shown in Fig. 3, we found there is room for improvement. With ResNet152, the produced heatmaps lack strong location sensitivity, occasionally missing large strokes. In the case of DenseNet121, likely due to numerous skip-connections, the heatmaps are focused but the regions of high intensity are too broad for detailed features such as SAH. When training DenseNet from scratch without the ImageNet pretrained weights (referred to as DenseNet* in Fig. 3), the heatmaps become less focused, indicating that pretraining on a large and diverse image set aids heatmap quality. For newer models like ResNet-RS and RDNet, the heatmaps are more detailed. Additional examples can be found in Fig. 6.

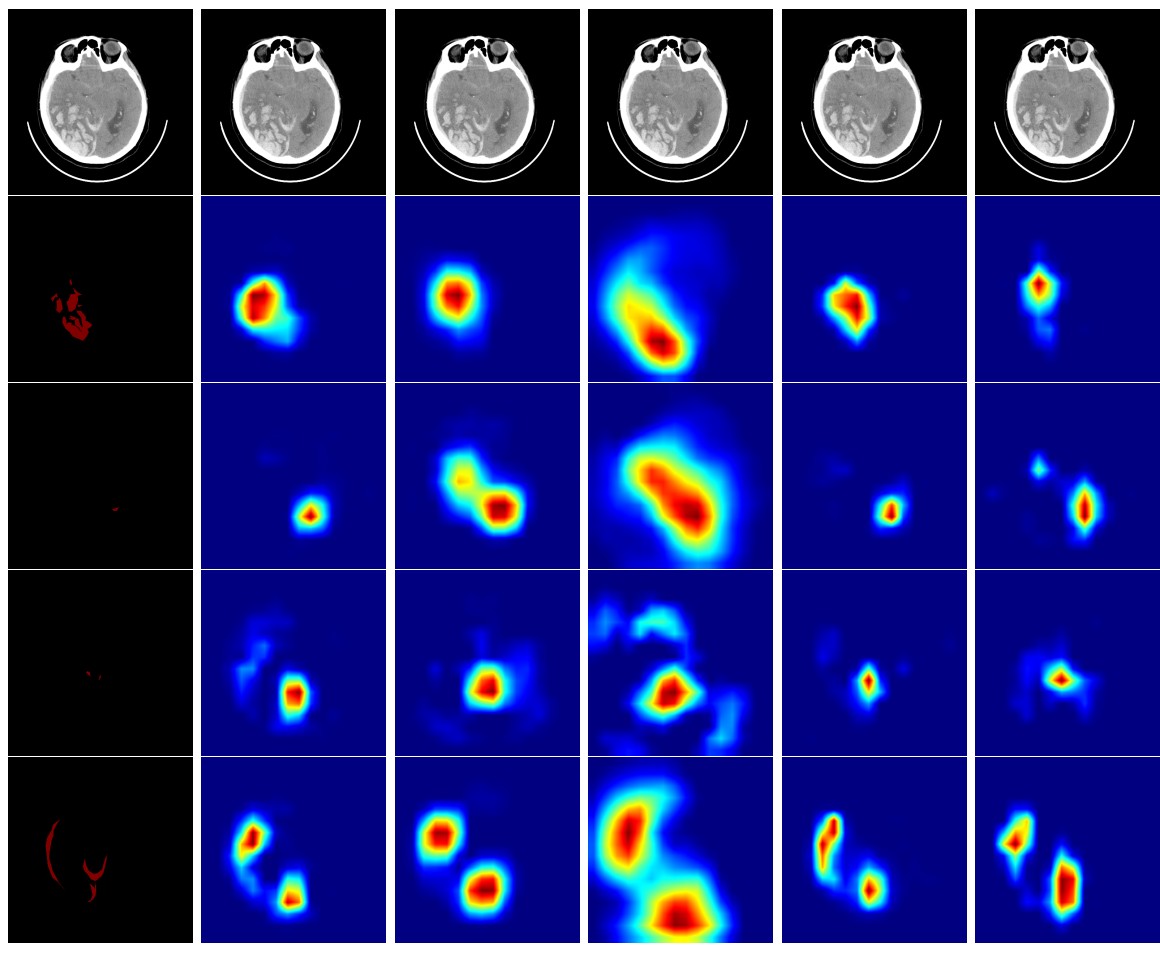

GT        ResNet152      DenseNet121  DenseNet121* ResNet-RS200      RDNet

Figure 2: ICH subtype heatmap generations selected from RSNA intracranial hemorrhage dataset (Flanders et al., 2020) . From the above to below: IPH, IVH, SAH, SDH. DenseNet* refers to DenseNet without pretrained model.

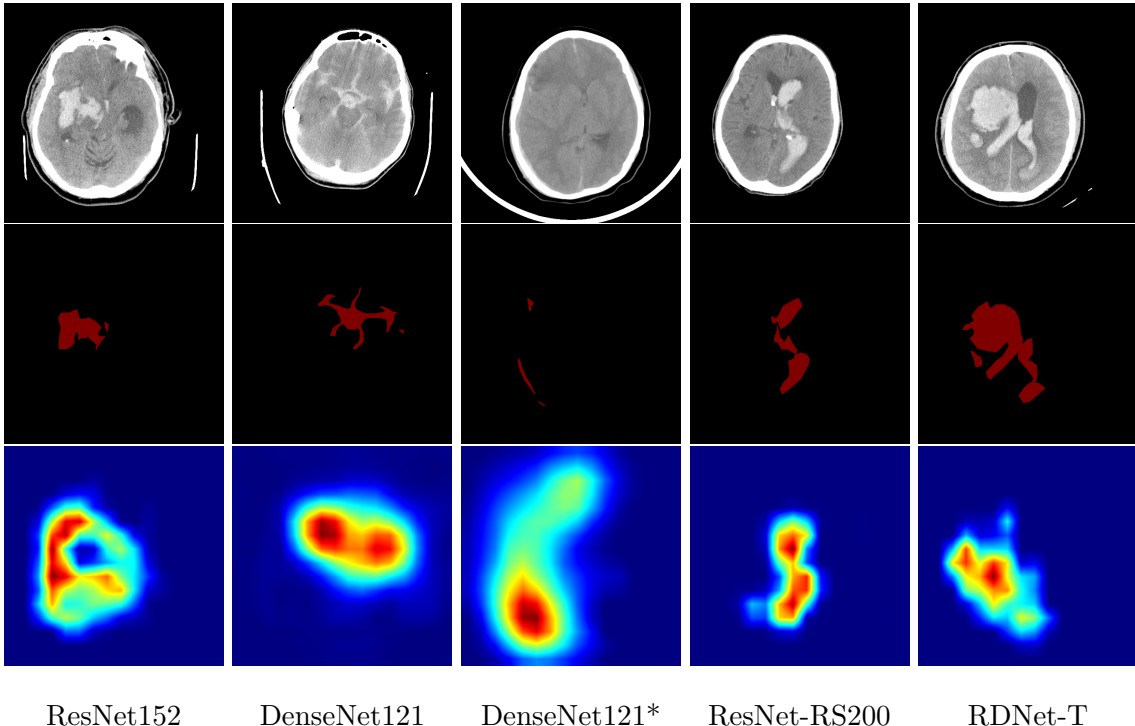

ResNet152  DenseNet121  DenseNet121*  ResNet-RS200  RDNet-T

Figure 3: Illustrations of heatmap generations. (*= no-pretrained model.)

### 3.2. Architecture

The main proposed architecture is illustrated in Fig. 1. It is a UNet-like architecture with encoder that is first pretrained on the target dataset. For the encoder backbone, we use RDNet-T (Kim et al., 2024), an updated version of DenseNet designed to enhance accuracy and scaling strategy compared to the original. Modern architectures like patchification stem, base block from ConvNeXt (Liu et al., 2022) are adopted. And the transition layer is redesigned and applied after every three blocks instead of only after each stage. Working on images with size $512 \times 512$, we added additional parameters at the end of each stage, corresponding to spatial dimensions of $128 \times 128, 64 \times 64, 32 \times 32$, and $16 \times 16$, with channel sizes of $256, 440, 744, 1040$, respectively.

For the decoder, each stage primarily includes three components: a convolution block, an upsampling block, and the final classification layer. The convolution block applies 1D convolution to convert the concatenated features into the desired channel dimension, followed sequentially by a Layer Normalization layer, a ReLU activation, and a squeeze-and-excitation block. The upsampling block first uses 1D convolution to adjust the channel dimension to match that of the previous stage, then applies 2D convolution to increase the number of channels by four times, and performs upsampling using the depth-to-space operation. A Layer Normalization layer is added at the end of the upsampling block. Lastly, the final classification layer produces multi-label scores, which are then fed into the LSE pooling layer. Heatmaps are generated just before the top LSE pooling layer. As discussed in Section 2.4, when patch-level predictions are trained with high accuracy, the resulting

heatmaps tend to be sharper compared to those produced by networks that use global average pooling layers.

### 3.3. Loss Function

Let $f_0, \ldots, f_3$ represent the inputs of the LSE pooling layer, and $l_0, \ldots, l_3$ the outputs following the LSE pooling layer, from above to below. Denote $gt$ the ground truth label. Then we have the classification loss

$$\text{CLoss} = \alpha_0 \text{BCE}(l_0, gt) + \alpha_1 \text{BCE}(l_1, gt) + \alpha_2 \text{BCE}(l_2, gt) + \alpha_3 \text{BCE}(l_3, gt), \tag{6}$$

where BCE represents the binary cross entropy loss function and $\alpha_0, \ldots, \alpha_3$ are weight coefficients. Additionally, to ensure consistency in heatmap generation at each level, a heatmap consistency loss is applied,

$$\text{HLoss} = \beta_0 \text{MSE}(\text{Pool}(f_0), f_1) + \beta_1 \text{MSE}(\text{Pool}(f_1), f_2) + \beta_2 \text{MSE}(\text{Pool}(f_2), f_3), \tag{7}$$

where MSE represents the mean square error loss, Pool refers to average pooling with a size of 2, and $\beta_0, \ldots, \beta_2$ are weight coefficients. The overall loss is the sum of the classification loss (CLoss) and the heatmap consistency loss (HLoss).

### 3.4. Weakly Supervised Quantization Algorithm

Based on the observations that the generated heatmaps for different disease subtypes are nearly mutually exclusive, our objective is to produce class-specific heatmaps for each category, subsequently determining the final predicted mask by employing the argmax function. Nevertheless, we have empirically observed that Grad-CAM tends to yield an excessive number of false positive results. In order to mitigate this concern, we introduce two distinct thresholds: class thresholds and mask thresholds. The class thresholds are selected based on the values that yield maximal Youden indices. We generate the class-specific heatmap solely when the corresponding output surpasses the designated class threshold for that specific category. Conversely, given our intent to retain only the regions highlighted in red in Fig. 2, therefore we set mask thresholds: heatmap values that fall below these mask thresholds are adjusted to zero. We use grid search to find the mask thresholds, and the details are provided in Appendix B.

## 4. Experiments

In this work, we validate our proposed approach for intracranial hemorrhage subtype classification. In 2019, the RSNA hosted a competition centered on classifying acute intracranial hemorrhage subtypes (Flanders et al., 2020), providing over 20,000 CT scans along with slice-level classification labels. The class annotations include five primary ICH subtypes: intraparenchymal hemorrhage (IPH), intraventricular hemorrhage (IVH), epidural hematoma (EDH), subdural hematoma (SDH), and subarachnoid hemorrhage (SAH). We randomly selected $2,734$ CT scans ($91,844$ slices) for training, and 89 CT scans ($3,000$ slices) for validation of subtype classification. The class thresholds used in Algorithm 1 were chosen

based on the threshold that maximized the difference between true positive rate and false positive rate on the validation set. For testing, we utilized the BHSD dataset (Wu et al., 2023), which contains 192 CT scans with subtype segmentation labels, as the test dataset.

For image input, we followed the approach used by (Kuo et al., 2019): all CT slices were adjusted with a window width of 130 and a window center of 25. We opted to stack three consecutive slices as input and use the segmentation mask of the middle slice as the output, instead of employing a three-dimensional model. To the best of our knowledge, using three consecutive slices is generally sufficient for accurately detecting and classifying subtype hemorrhages in almost all cases, except for a small number of instances that require distinguishing between EDH and SDH. We utilize RDNet as the framework to create the heatmap through guided backpropagation. For the integrated gradient, we apply the class threshold (in Algorithm 1) from RDNet and employ grid search to identify the appropriate mask thresholds. To evaluate the Dice coefficients at the instance level, we choose instances for each subtype with a volume exceeding 3000 (to minimize false negatives) and compute the binary Dice coefficients. There are 90 cases for IPH, 40 cases for IVH, 45 cases for SAH, 45 cases for SDH, and 14 cases for EDH. All experiments were carried out using a single Nvidia V100 GPU. For models that utilized pretrained checkpoints, we applied SGD with a step learning rate schedule over 20 epochs. For models without pretrained checkpoints, we employed Adam optimizer with a learning rate of $1e - 4$ for 30 epochs. Detailed hyperparameter settings can be found in Appendix C.

## 5. Results

The AUROCs on the validation dataset are shown in Table 1. Although our method achieves the highest performance, we observed that all models yield similar results. However, as shown in Table 2 and Table 3, our method significantly outperforms all other models across all disease subtypes. The heatmaps generated by our proposed method are illustrated in Fig. 4 and Fig. 5. When compared to the ground truth labels and the heatmaps produced by other network architectures shown in Fig. 3, our method clearly performs better in accurately capturing the shapes and locations of hemorrhages.

The fifth row of Fig. 5. displays the heatmaps produced by guided backpropagation. While many of these heatmaps seem reasonable, their performance is surprisingly inferior to that of RDNet alone. This can be possibly due to too many noises in the heatmaps. On the other hand, , the heatmaps produced by integrated gradients are shown in the final row of Fig. 5. We noticed that the heatmaps are focused on the skulls of the brains and other areas not related to the class, and the performance levels do not match those of the other baseline models. We want to highlight that the heatmap consistency loss described in Equation 7 is essential. Without this regularization during training, as shown in Fig. 5, the resulting heatmaps tend to have a grid-like pattern and are less focused on the stroke areas. Therefore the results demonstrated in Table 1 considerably deteriorate.

For ICH strokes, all types except IPH are irregular, and SAH lesions are typically small and difficult to capture with low-resolution heatmaps. On the other hand, because we generate heatmaps in a latent space of size $128 \times 128$, our approach can better approximate irregular or small, fine-grained shapes compared to other methods.

Table 1: AUROCs on the validation dataset. (*= no-pretrained model.)

| Model | IPH | IVH | SAH | SDH | EDH | AVG |
|-------|-----|-----|-----|-----|-----|-----|
| ResNet152 | 0.970 | 0.997 | 0.949 | 0.923 | 0.941 | 0.956 |
| DenseNet121 | **0.974** | 0.997 | 0.957 | 0.918 | 0.954 | 0.960 |
| DenseNet121* | 0.973 | 0.997 | 0.960 | 0.915 | 0.963 | 0.961 |
| ResNetRS | 0.971 | 0.996 | 0.956 | 0.926 | 0.956 | 0.961 |
| RDNet | 0.973 | 0.996 | 0.948 | 0.916 | **0.975** | 0.962 |
| Ours | 0.973 | **0.998** | **0.959** | **0.934** | 0.969 | **0.967** |

Table 2: DICE coefficients on the BHSD dataset, using the class thresholds of the models to generate heatmaps. (*= no-pretrained model.)

| Model | Normal | IPH | IVH | SAH | SDH | EDH | AVG |
|-------|--------|-----|-----|-----|-----|-----|-----|
| ResNet152 | 0.992 | 0.242 | 0.093 | 0.056 | 0.075 | 0.060 | 0.253 |
| DenseNet121 | 0.990 | 0.256 | 0.113 | 0.060 | 0.070 | 0.078 | 0.261 |
| DenseNet121* | 0.989 | 0.260 | 0.132 | 0.047 | 0.076 | 0.054 | 0.260 |
| ResNetRS | 0.996 | 0.445 | 0.182 | 0.138 | 0.144 | 0.158 | 0.344 |
| RDNet | 0.996 | 0.380 | 0.211 | 0.108 | 0.133 | 0.141 | 0.328 |
| Guided Gradient | 0.991 | 0.110 | 0.232 | 0.126 | 0.106 | 0.111 | 0.279 |
| Integrated Gradient | 0.914 | 0.004 | 0.012 | 0.007 | 0.002 | 0.011 | 0.158 |
| Ours (no HLoss) | 0.990 | 0.142 | 0.388 | 0.137 | 0.082 | 0.112 | 0.309 |
| Ours | **0.997** | **0.551** | **0.387** | **0.225** | **0.245** | **0.356** | **0.460** |

Table 3: Instance DICE coefficients on the BHSD dataset, using the class thresholds of the models to generate heatmaps. (*= no-pretrained model.)

| Model | IPH | IVH | SAH | SDH | EDH |
|-------|-----|-----|-----|-----|-----|
| ResNet152 | $0.248 \pm 0.14$ | $0120 \pm 0.06$ | $0.077 \pm 0.05$ | $0.115 \pm 0.09$ | $0.255 \pm 0.16$ |
| DenseNet121 | $0.286 \pm 0.17$ | $0.143 \pm 0.08$ | $0.085 \pm 0.05$ | $0.110 \pm 0.09$ | $0.169 \pm 0.13$ |
| DenseNet121* | $0.258 \pm 0.16$ | $0.126 \pm 0.07$ | $0.053 \pm 0.03$ | $0.086 \pm 0.08$ | $0.121 \pm 0.10$ |
| ResNetRS | $0.414 \pm 0.18$ | $0.213 \pm 0.07$ | $0.119 \pm 0.07$ | $0.185 \pm 0.12$ | $0.272 \pm 0.15$ |
| RDNet | $0.354 \pm 0.16$ | $0.227 \pm 0.08$ | $0.130 \pm 0.07$ | $0.178 \pm 0.13$ | $0.255 \pm 0.16$ |
| Ours | $\mathbf{0.514 \pm 0.17}$ | $\mathbf{0.398 \pm 0.10}$ | $\mathbf{0.204 \pm 0.11}$ | $\mathbf{0.228 \pm 0.12}$ | $\mathbf{0.366 \pm 0.19}$ |

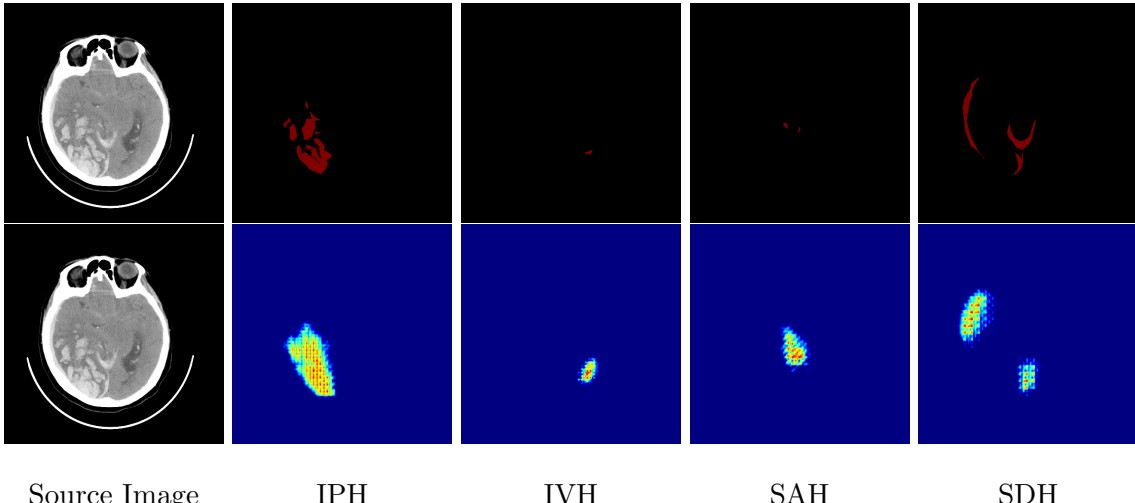

| Source Image | IPH | IVH | SAH | SDH |

Figure 4: Illustrations of subtype heatmaps generations by our proposed method. From above to below: ground truth labels, heatmaps generated by our proposed method.

## 6. Conclusion

Creating detailed annotations is a labor-intensive process. Nevertheless, quantification is essential in certain clinical contexts. In this research, we explored weakly supervised segmentation of intracerebral hemorrhage (ICH) subtypes. Surprisingly, we found that while all models showed similar classification accuracy, the quality of the heatmaps they generated varied significantly. Although every model correctly identified the subtype location, advanced models with pretrained checkpoints produced more detailed heatmaps closely matched subtype stroke segmentation masks.

However, all heatmaps were generated from relatively low-resolution feature maps, for example, $16 \times 16$. To harness the full potential of these advanced models and pretrained checkpoints, we developed a decoder for these pretrained models and retrained them on higher-resolution feature maps using the Log-Sum-Exp (LSE) pooling layer. We carefully enforced consistency across latent spaces at each level and ultimately generated heatmaps from the latent space with the highest resolution. Our findings indicate that even without segmentation labels, well-trained classifiers may possess some capacity for quantification. We believe our approach can be useful in situations where training a segmentation model is costly. We plan to systematically explore this direction in future work.

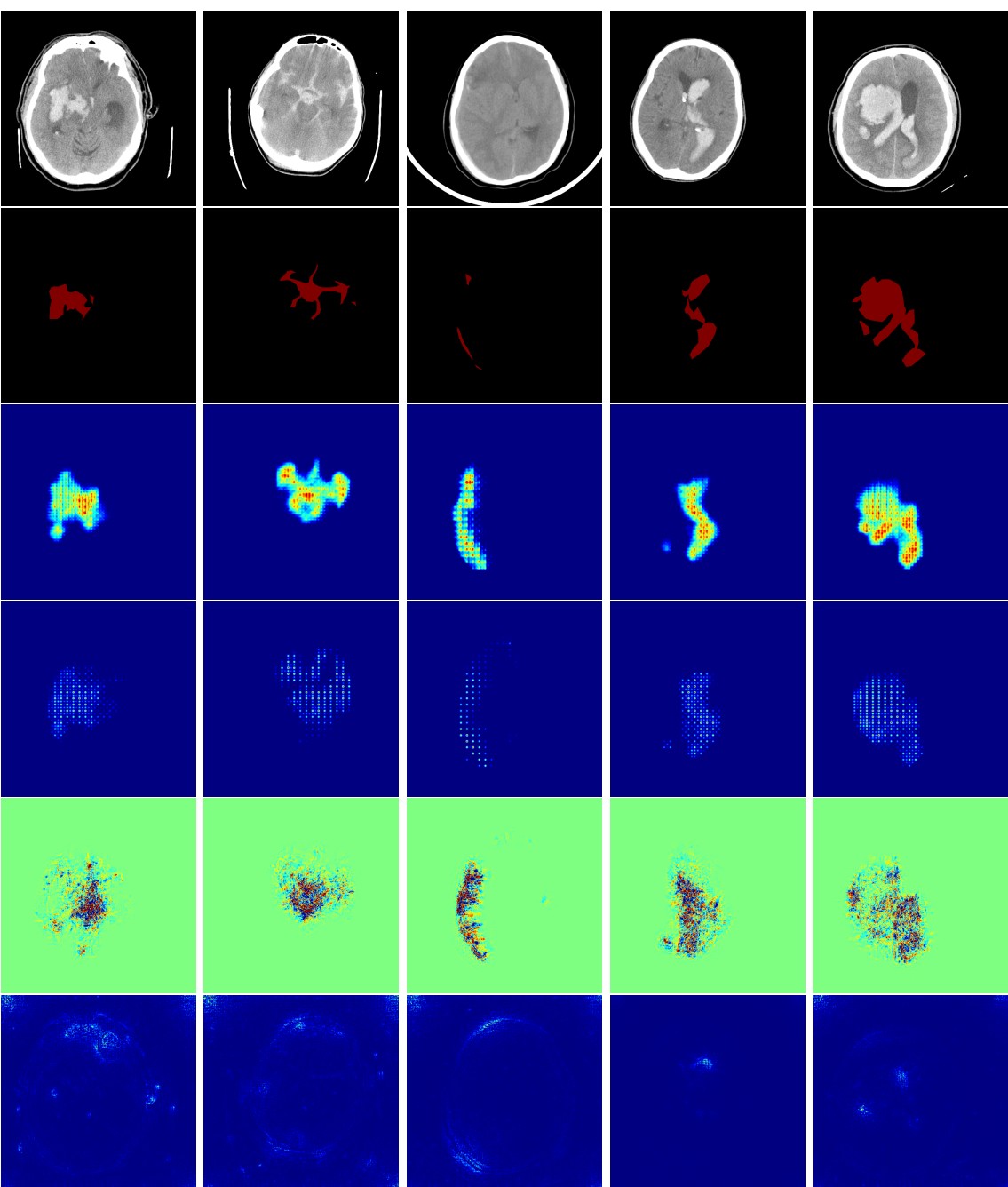

Figure 5: Illustrations of the heatmap generations of ICH by our proposed method. From above to below: original images, ground truths, heatmap generated by our proposed method, heatmap generated by our proposed method without the heatmap consistency loss (HLoss), heatmaps generated by applying guided backpropagation, heatmap generated by integrated gradients.

## Acknowledgments

Both authors acknowledge funding from the Center for Artificial Intelligence in Medicine at Chang Gung Memorial Hospital, via grant agreements no. CLRPG3H0016 and no. CORPG3L0463

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

## Appendix A.  Pseudo Code for the Weakly Supervised Quantization

The pseudo-code of our main algorithm is summarized in Algorithm 1.

## Appendix B. Performance for Pure Visualization

We aim to demonstrate that when the ground truth labels are known (i.e., after eliminating all false positive and false negative cases), although our model achieves the highest performance, advanced models such as ResNetRS and RDNet are also suitable for ICH quantification. First, we described how we selected our mask thresholds. We rescaled all heatmaps to the range $[0, 255]$, and then applied grid search on thresholds $m = 25, 50, 75, 100, 125, 150, 175, 200$ to evaluate subtype performance. As shown in Fig. 2, we empirically observed that these

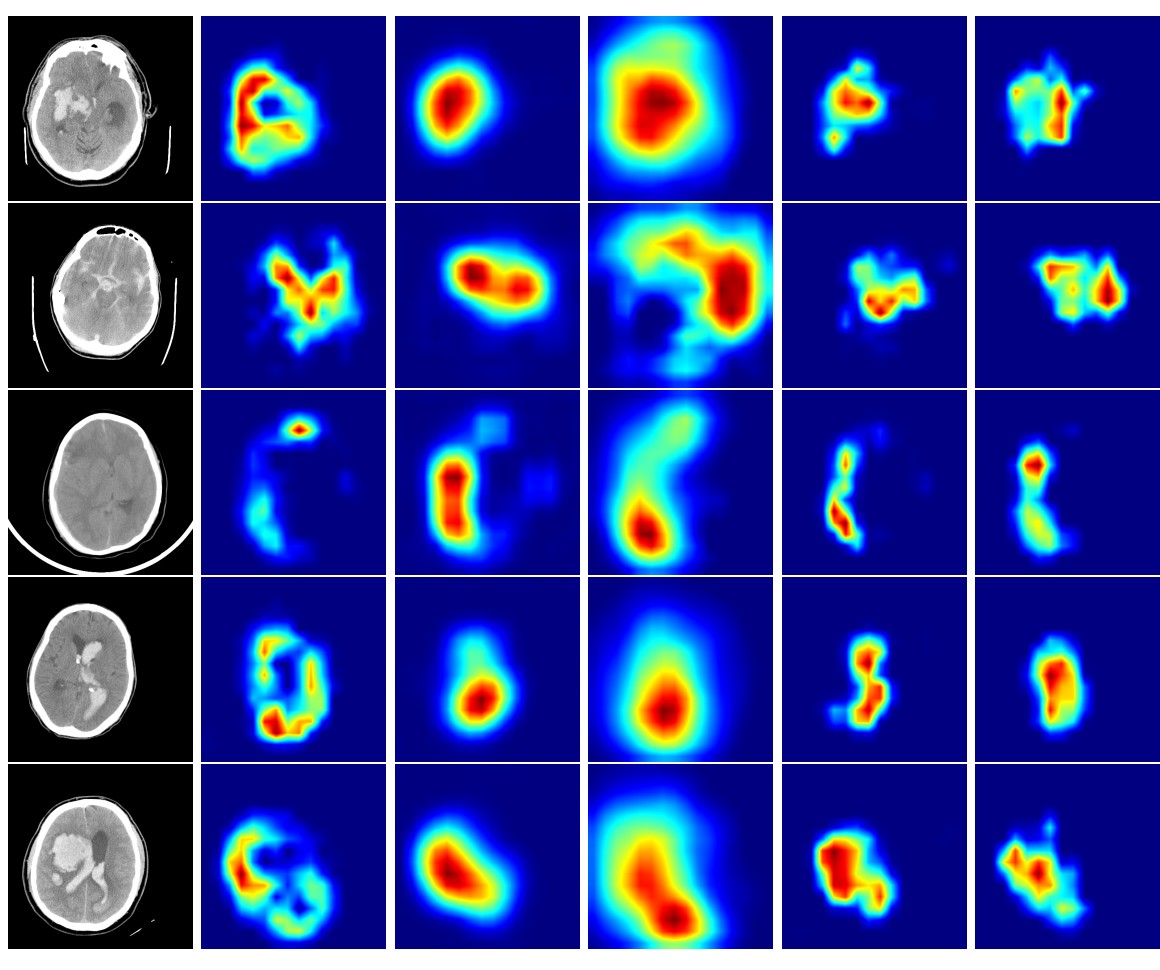

| Origin | ResNet152 | DenseNet121 | DenseNet121* | ResNet-RS200 | RDNet |

Figure 6: More examples for different network architectures.

---

**Algorithm 1:** Pseudocode for Subtype Segmentation Mask Generation

---

**Input:** Class thresholds $c_{\text{IPH}}, \ldots, c_{\text{EDH}}$. Mask thresholds $m_{\text{IPH}}, \ldots, m_{\text{EDH}}$. Neural
        network $N$, image $I$, image size $(h, w)$.

**Output:** Segmentation masks $M$

$M \leftarrow$ np.zeros((6, h, w))

**for** $i \leftarrow$ *ICH subtypes* **do**
    **if** $N(I)[i] > c_i$ **then**
        $M[i] \leftarrow$ np.where(Grad-CAM$(I) > m_i$, Grad-CAM$(I), 0$);

**end**

$M \leftarrow$ np.argmax(M, 0)

**return** $M$

---

Table 4: Selected mask thresholds.

| Model | $m_{IPH}$ | $m_{IVH}$ | $m_{SAH}$ | $m_{SDH}$ | $m_{EDH}$ |
|---|---|---|---|---|---|
| ResNet152 | 150 | 150 | 175 | 150 | 175 |
| DenseNet121 | 175 | 200 | 200 | 175 | 200 |
| DenseNet121* | 200 | 200 | 200 | 200 | 200 |
| ResNetRS | 150 | 175 | 150 | 150 | 175 |
| RDNet | 175 | 200 | 175 | 175 | 200 |
| Ours | 75 | 125 | 125 | 75 | 25 |

subtype heatmaps are mutually exclusive. Therefore, we tested the subtype thresholds jointly. The final mask thresholds are summarized in Table 4.

The final results are presented in Table 5. We observed that ResNetRS, RDNet, and our model all achieved a Dice coefficient above 0.5 for IPH. We believe these results are sufficiently accurate for some situations where clinicians need to include quantified information in radiology reports, but was previously performed using less precise estimation methods.

Table 5: DICE coefficients on the BHSD dataset, using the ground truth labels to generate heatmaps.

| Model | Normal | IPH | IVH | SAH | SDH | EDH | AVG |
|---|---|---|---|---|---|---|---|
| ResNet152 | 0.996 | 0.359 | 0.116 | 0.105 | 0.158 | 0.281 | 0.336 |
| DenseNet121 | 0.995 | 0.425 | 0.162 | 0.122 | 0.166 | 0.303 | 0.362 |
| DenseNet121* | 0.993 | 0.286 | 0.151 | 0.084 | 0.131 | 0.205 | 0.308 |
| ResNetRS | 0.998 | 0.552 | 0.245 | 0.193 | 0.243 | 0.425 | 0.443 |
| RDNet | 0.997 | 0.502 | 0.264 | 0.173 | 0.234 | 0.373 | 0.424 |
| Ours | 0.998 | 0.607 | 0.424 | 0.254 | 0.281 | 0.445 | 0.502 |

Table 6: DICE coefficients on the BHSD dataset, using the ground truth labels to generate heatmaps and with all mask thresholds equal to 125.

| Model | Normal | IPH | IVH | SAH | SDH | EDH | AVG |
|---|---|---|---|---|---|---|---|
| ResNet152 | 0.993 | 0.354 | 0.112 | 0.094 | 0.161 | 0.242 | 0.326 |
| DenseNet121 | 0.990 | 0.352 | 0.106 | 0.854 | 0.140 | 0.186 | 0.310 |
| DenseNet121* | 0.978 | 0.169 | 0.057 | 0.051 | 0.085 | 0.123 | 0.244 |
| ResNetRS | 0.996 | 0.539 | 0.211 | 0.183 | 0.249 | 0.361 | 0.423 |
| RDNet | 0.994 | 0.468 | 0.182 | 0.143 | 0.213 | 0.295 | 0.383 |
| Ours | 0.998 | 0.576 | 0.424 | 0.254 | 0.236 | 0.255 | 0.457 |

Table 7: DICE coefficients on the BHSD dataset, using the ground truth labels to generate heatmaps and with all mask thresholds equal to 75.

| Model | Normal | IPH | IVH | SAH | SDH | EDH | AVG |
|---|---|---|---|---|---|---|---|
| ResNet152 | 0.988 | 0.293 | 0.089 | 0.071 | 0.138 | 0.171 | 0.291 |
| DenseNet121 | 0.983 | 0.263 | 0.078 | 0.062 | 0.108 | 0.128 | 0.270 |
| DenseNet121* | 0.960 | 0.100 | 0.035 | 0.035 | 0.062 | 0.079 | 0.211 |
| ResNetRS | 0.993 | 0.450 | 0.158 | 0.140 | 0.211 | 0.265 | 0.369 |
| RDNet | 0.990 | 0.377 | 0.126 | 0.105 | 0.166 | 0.212 | 0.329 |
| Ours | 0.997 | 0.607 | 0.366 | 0.207 | 0.281 | 0.381 | 0.473 |

Table 8: DICE coefficients on the BHSD dataset, using the ground truth labels to generate heatmaps and with all mask thresholds equal to 25.

| Model | Normal | IPH | IVH | SAH | SDH | EDH | AVG |
|---|---|---|---|---|---|---|---|
| ResNet152 | 0.973 | 0.197 | 0.057 | 0.042 | 0.083 | 0.099 | 0.242 |
| DenseNet121 | 0.967 | 0.167 | 0.047 | 0.037 | 0.071 | 0.070 | 0.226 |
| DenseNet121* | 0.924 | 0.053 | 0.021 | 0.022 | 0.041 | 0.043 | 0.184 |
| ResNetRS | 0.986 | 0.322 | 0.097 | 0.088 | 0.143 | 0.161 | 0.299 |
| RDNet | 0.980 | 0.262 | 0.076 | 0.064 | 0.101 | 0.125 | 0.268 |
| Ours | 0.996 | 0.577 | 0.293 | 0.168 | 0.253 | 0.445 | 0.455 |

Table 9: DICE coefficients with different hyperparameters. Double refers to we double the parameters $\beta_0, \beta_1$, and $\beta_2$, and similarly Half means we halve $\beta_0, \beta_1$, and $\beta_2$.

| Model | Normal | IPH | IVH | SAH | SDH | EDH | AVG |
|-------|--------|-----|-----|-----|-----|-----|-----|
| Half | 0.996 | 0.530 | 0.294 | 0.206 | 0.243 | 0.291 | 0.427 |
| Double | 0.996 | 0.539 | 0.286 | 0.199 | 0.222 | 0.265 | 0.418 |

## Appendix C. Hyperparameter Setting

For the hyperparameters in Equations 6 and 7, in this work, we set $\alpha_0 = 2, \alpha_1 = 0.5, \alpha_2 = 0.25, \alpha_3 = 0.125, \beta_0 = 0.5, \beta_1 = 0.25$, and $\beta_2 = 0.125$. Additionally, the scaling factor $r$ in the LSE pooling layer (Equation 5) is set to 3.

To demonstrate our method is robust to hyperparameter tuning, we try to double/halve $\beta_0, \beta_1$, and $\beta_2$ and retrain the models for 10 epochs. For simplicity, we use the the same mask threshold of our default setting in Table 4, thereby potentially leading to suboptimal outcomes. The performances are presented in Table 9.

## Appendix D. Results for different hyperparameters

We discovered that mask thresholds play a key role in the final heatmap quantization. Some of the results are presented in Table 6, Table 7, and Table 8.

## Appendix E. Heuristics about the Heatmap Consistency Loss

In this section we would like to provide an intuitive explanation for the heatmap consistency loss in Equation 7. Following the notation in Section 3.3, locally the loss will be

$$f_{i,j}^{k,1} + f_{i,j}^{k,2} + f_{i,j}^{k,3} + f_{i,j}^{k,4} = f_{i,j}^{k+1}, \tag{8}$$

where $f_{i,j}^{k+1}$ is the value of $f^{k+1}$ at spatial location $(i, j)$, and $f_{i,j}^{k,1}, \ldots, f_{i,j}^{k,4}$ are values of $f^k$ lying above $(i, j)$. Now we assume $f_{i,j}^{k,l} >= f_{min}$, for a designed minimum $f_{min}$. For $r > 0$, we rewrite the LSE function in Equation 4 as

$$\log(\Sigma_{i,j}(\exp(r \cdot f_{i,j}^{k,1}) + \exp(r \cdot f_{i,j}^{k,2}) + \exp(r \cdot f_{i,j}^{k,3}) + \exp(r \cdot f_{i,j}^{k,4})) = \log(\Sigma_{i,j}L_{i,j}^k). \tag{9}$$

On the other hand, from the inequality of arithmetic and geometric means,

$$\frac{L_{i,j}^k}{4} \geq (\exp(r \cdot (f_{i,j}^{k,1} + f_{i,j}^{k,2} + f_{i,j}^{k,3} + f_{i,j}^{k,4}))^{\frac{1}{4}} = \exp(\frac{r \cdot f_{i,j}^{k+1}}{4}), \tag{10}$$

and the unique minimum is achieved when $f_{i,j}^{k,1} = f_{i,j}^{k,2} = f_{i,j}^{k,3} = f_{i,j}^{k,4}$. On the other hand, to find the maximum value, without loss of generalization, we assume $f_{i,j}^{k,1} > f_{i,j}^{k,2} \geq f_{i,j}^{k,3} \geq f_{i,j}^{k,4}$, then we rewrite $f_{i,j}^{k,4} = f_{i,j}^{k+1} - (f_{i,j}^{k,1} + f_{i,j}^{k,2} + f_{i,j}^{k,3})$, then

$$\frac{\partial L_{i,j}^k}{\partial f_{i,j}^{k,1}} = r * (\exp(r \cdot f_{i,j}^{k,1}) - \exp(r \cdot f_{i,j}^{k,4})) > 0, \tag{11}$$

which means that we can increase $L_{i,j}^k$ by increasing $f_{i,j}^{k,1}$ until $f_{i,j}^{k,4}$ achieves the designed lower bound $f_{min}$. Recursively following the argument, we find the maximum value is achieved when

$$f_{i,j}^{k,1} = f_{i,j}^{k+1} - 3 \cdot f_{min}, \quad f_{i,j}^{k,2} = f_{i,j}^{k,3} = f_{i,j}^{k,4} = f_{min}. \tag{12}$$

In sum, if the spatial location $(i, j)$ belongs to the pathological region, then only one of $f_{i,j}^{k,l}$ achieves the extreme value while the remaining ones tends to achieve the minimum value. On the other hand, if the spatial location $(i, j)$ belongs to the normal region, all the four values tends to achieve the $\frac{f_{i,j}^{k+1}}{4}$, hence the heatmap will be more fine-grained in the outputs of the upper levels.

