# OpenReview forum: "Beyond Classification: Elaborating Network Predictions for Better Weakly Supervised Quantization"
_MIDL.io/2026/Conference — MIDL 2026 Poster_

### Official Review · Reviewer_ucPs · 2025-12-27

**Confidence:** 4
**Preliminary Rating:** 2
**Final Rating:** 3

**Summary:**

This work leverages pretrained CNN models and a Log-Sum-Exp pooling layer to develop a U-Net–like architecture for weakly supervised segmentation. The proposed method was evaluated on the ICH dataset for quantifying intracranial hemorrhage subtypes and demonstrated improved localization performance compared to Grad-CAM, while maintaining comparable classification performance.

**Strengths:**

- The method is evaluated across several CNN backbone architectures and two different datasets were used, one for training and the other for evaluation, which helps account for data variability
- Heatmap consistency is enforced across the latent representations at each level, where heatmaps are generated from the latent space with the highest resolution (128x128)
- The authors demonstrate that leveraging pretrained models as encoders, combined with an appropriate decoder, enables effective weakly supervised segmentation, thereby reducing annotation costs.
- the proposed method outperform the naive GradCAM in term of DICE score while preserving classification performance
- The method achieves higher Dice scores than standard Grad-CAM without compromising classification performance

**Weaknesses:**

- The manuscript suffers from poor organization and writing quality, with results and parts of the discussion presented in the introduction before the methodology section
- The choice of CNN backbone architectures are not well motivated.
- The architecture section does not clearly explain the encoder design, and the corresponding figure is hard to interpret without sufficient textual descriptions
- The resulting loss function is overly complex, involving numerous hyperparameters, which makes it difficult to adapt the method to new tasks
- The code is not publicly available, and coupled with the insufficient methodological description, this makes the study difficult to reproduce

**Detailed Comments:**

- The abbreviation "ROI" is used in the abstract without being defined
- It is unclear how the use of heatmaps contributes to lowering the false negative rate in diagnoses, as claimed in the introduction
- The authors should distinguish between gradient-free and gradient-based methods in the first paragraph of the introduction. CAM is incorrectly described as a gradient-based method, whereas it is not.
- Figure 2 is difficult to interpret due to the lack of clear ground truth annotations. Furthermore, the models are evaluated on different images, which complicates the analysis of localization sensitivity. It is also unclear which dataset was used for Figures 1 and 2
- The related work section is limited and does not cover recent works in the field of medical imaging
- Equation 1 is unclear, as each neuron in the fully connected layer should contribute to the output score for every class
- The term “quantization” in the title may be misleading, as the paper does not provide sufficient motivation or explanation for its use

**Justification Of Final Rating:**

While the paper addresses a relevant problem (namely weakly supervised segmentation) it suffers from several limitations that weaken the overall contribution. In particular:

- **Writing quality**: The manuscript would benefit from significant improvements in clarity and organization.
- **Hyperparameter selection**: The proposed method relies heavily on numerous hyperparameters, including loss-function weights ($\alpha_1$, $\alpha_2$, $\alpha_3$, $\beta_1$, $\beta_2$, $\beta_3$ ​
) intrinsic to the model, as well as posthoc evaluation parameters (e.g., thresholding). Without validation across multiple tasks, the paper does not provide clear guidelines or a principled strategy for selecting or tuning these hyperparameters.
- **Qualitative analysis**: The qualitative visual evaluation does not clearly demonstrate the advantages of the proposed method over the baselines. In addition, the visualizations are not presented using a unified color map, which hampers fair and intuitive comparison.
- **Connection to explainability**: Although the authors emphasize that the task is weakly supervised segmentation rather than explainability, the distinction between these two concepts is not clearly articulated. As a result, relevant self-explainable models that generate explanations from image-level labels should have been considered as potential baselines.

Given these concerns, I would recommend rejection, despite the method itself appearing promising.

**Justification Of The Preliminary Rating:**

The paper is poorly organized, and although the proposed method is promising, it was evaluated on a single dataset and modality, which limits its generalizability. Moreover, the method was not compared to appropriate post-hoc approaches that provide high-resolution feature maps. The absence of publicly available code, combined with insufficient methodological detail, further restricts reproducibility. Finally, the loss function is overly complex, containing six hyperparameters, which makes it difficult to fine-tune for other tasks

**Questions To Address In The Rebuttal:**

See weaknessess
- It is unclear why a classification layer is applied at each stage of the decoder, rather than using a single classification layer at the final decoder stage
- The paper does not provide information on how the coefficients of the loss function are selected or their impact on model performance
- It is unclear why a 128×128 heatmap is generated instead of producing a heatmap that directly matches the input size, which would avoid the need for hard upsampling
- Backpropagation-based methods, such as Integrated Gradients and Guided Backpropagation, provide high-resolution feature maps that correspond to the input size and overcome the low-resolution limitations of Grad-CAM. It is unclear how the proposed method compares to these approaches or whether they could be integrated into the current framework.
- The proposed approach appears similar to that described in [ https://arxiv.org/abs/2505.17748 ]. The authors should clarify how their method differs from or improves upon this prior work

---

> ### Author Response · Authors · 2026-01-24
> **Manuscript is revised, code is available at  https://github.com/chihchiehchen/Beyond-Classification-Elaborating-Network-Predictions-for-Better-Weakly-Supervised-Quantization**
>
> Dear Reviewer ucPs,
>
> Thanks for your informative suggestions. We summarize our responds as follows:
>
> * The absence of publicly available code, combined with insufficient methodological detail, further restricts reproducibility.
>
> We respectfully disagree with Reviewer. The code has previously been available (since we submitted the article) on our github page, as mentioned in the "Reproducibility:" column above.
>
> * It is unclear why a classification layer is applied at each stage of the decoder, rather than using a single classification layer at the final decoder stage
>
>   We use the concept of deep supervision in the segmentation task. In our scenario, because we do not have ground truth segmentation labels, we use log-sum-exp map as our final output.
>
> * The paper does not provide information on how the coefficients of the loss function are selected or their impact on model performance
>
> We completely follow the coefficients, just as we did in deep supervision. Specifically, the coefficients of lower layers will be one-half of those of upper levels.  Due to time constraints, we did not complete all of the trials; nonetheless,  partial results are available in Appendix G.
>
> * Backpropagation-based methods, such as Integrated Gradients and Guided Backpropagation, provide high-resolution feature maps that correspond to the input size and overcome the low-resolution limitations of Grad-CAM.
>
> Thank you for the reviewer's idea; we now include these two approaches as baselines. For the integrated gradient, we use the official github page.  The results can be found in Figure 5 and Table 2. However, the outcomes are worse than Grad-CAM.
>
> * The proposed approach appears similar to that described in [ https://arxiv.org/abs/2505.17748 ].
>
> Thank you for explanation. We included it as a reference in Section 2.2 (highlighted in yellow). However, I believe the purposes of this work and ours are very different. While this work aims to get interpretable insights into the classifier, our focus is on weakly supervised segmentation and volume estimation. We apologize, but the pictures provided in this paper do not show how to perform weakly supervised segmentation.
>
> * It is unclear why a 128×128 heatmap is generated instead of producing a heatmap that directly matches the input size, which would avoid the need for hard upsampling
>
> We apologize, but we do not claim that upsampling is the only approach to obtain fine-grained heatmaps. We performed upsampling and demonstrated some quantitative results. It is also achievable using alternative ways. As the reviewer noted, Integrated Gradients and Guided Backpropagation are other viable options. However, when we tried it, it didn't work well.
>
> * The abbreviation "ROI" is used in the abstract without being defined
>
> Thanks for the reminder. We have now corrected it.
>
> * it is unclear how using heatmap can lower the false negative rate
>
> We apologize, but what we are asserting is that "by utilizing heatmaps or bounding boxes," which refers to any localization approach.
>
> * The authors should distinguish between gradient-free and gradient-based methods
>
> We respectfully disagree with Reviewer: this is not our primary objective. In the first paragraph of Section 1, we include some sentences to illustrate our motivations (highlighted in yellow). Our primary focus is on ICH subtype segmentation.
>
> * CAM is incorrectly described as a gradient-based method
>
> We apologize, but we did not claim that. We assert that Grad-CAM is equal to CAM when the final activation layer is the average pooling layer.
>
> * It is also unclear which dataset was used for Figures 1 and 2
>
> It's from RSNA dataset.
>
> * The related work section is limited
>
> Now we included more references in Section 2.3.
>
> * Equation 1 is unclear, as each neuron in the fully connected layer should contribute to the output score for every class
>
> The superscrips "c" in Equation 1 represents the target class.
>
> * The term “quantization” in the title may be misleading
>
> We are working on poorly supervised segmentation and volume estimation, which is a type of quantization.
>
> * The manuscript suffers from poor organization
>
> We relocate the technical content to section 3.1.
>
>  * The choice of CNN backbone architectures are not well motivated
>
> In this paper, we primarily investigate ResNet and DenseNet. RDNet is an updated version of DenseNet that performs significantly better on the ImageNet dataset, that is the reason we use it.

---

> > ### Comment · Reviewer_ucPs · 2026-01-28
> > **Despite the authors’ efforts to improve the manuscript, several points remain unclear.**
> >
> > Thank you for addressing my comments. Despite these efforts, some points remain unclear, notably the following:
> >
> > - **It is unclear why a classification layer is applied at each stage of the decoder, rather than using a single classification layer at the final decoder stage** This choice is not sufficiently motivated in the paper, and an ablation study would have been appropriate to support this architectural decision.
> >
> > - **The paper does not provide information on how the coefficients of the loss function are selected or their impact on model performance.** Unfortunately, Appendix G does not exist in the current version of the paper. In addition, the concept of deep supervision mentioned here does not appear to be introduced or discussed in the manuscript. Moreover, there is no guidance on how these hyperparameters could be selected for other tasks, and the method has only been evaluated on a single task.
> >
> > - **Backpropagation-based methods, such as Integrated Gradients and Guided Backpropagation, provide high-resolution feature maps that correspond to the input size and overcome the low-resolution limitations of Grad-CAM.** The visualizations are not presented using a unified color map, which hinders direct visual comparison. In addition, Guided Backpropagation appears to perform well qualitatively in the visual results (Fig. 5), but this is not reflected in the quantitative metrics (Tab. 2), and this discrepancy is not adequately discussed.
> >
> > - **The proposed approach appears similar to that described in [ https://arxiv.org/abs/2505.17748 ].** This work also aims to produce weakly supervised explanation maps (which can be viewed as weakly supervised segmentation maps) for explaining CNN classifiers, and could therefore serve as a solid baseline for this study.
> >
> > - **It is unclear why a 128×128 heatmap is generated instead of producing a heatmap that directly matches the input size, which would avoid the need for hard upsampling.** The initial question was not about comparisons with Integrated Gradients or Guided Backpropagation, but rather about avoiding low-resolution outputs (128×128) by constraining the architecture to produce explanations at the original input resolution, since upsampling may introduce noise.
> >
> > The method appears overly mechanistic, as there is no clear or principled procedure for selecting the optimal setup, ranging from loss-function hyperparameters to mask thresholding choices.
> >
> > Despite efforts to improve the paper, the writing quality still requires improvement.

---

> ### Author Response · Authors · 2026-01-29
> **Hyperparameters can be found in appendix C; we apologize for mistyping.**
>
> Dear Reviewer ucPs,
>
> Thank you for your responses and for giving us the opportunity to improve our content. We summarized our clarifications as follows:
>
> * It is unclear why a classification layer is applied at each stage of the decoder, rather than using a single classification layer at the final decoder stage.
>
> We partially agree with the reviewer. Table 2 and Figure 5 show that performance degrades significantly even without HLoss regularization. However, we all believe that further experimentation will make the work more compact. We will present additional findings before the end of the discussion period.
>
> * The paper does not provide information on how the coefficients of the loss function are selected or their impact on model performance.
>
>   We apologize for the typos: it is in Appendix C. All the hyperparameters we use in this work are included.
>
>   |Model | Normal| IPH | IVH  | SAH | SDH | EDH|AVG|
>   |------|------|------| ------|------|------| ------|------|
>   | Half | 0.996 | 0.530 | 0.294 | 0.206 | 0.243 | 0.291 | 0.427 |
>   | Double | 0.996 | 0.539 | 0.286 | 0.199 | 0.222 | 0.265 | 0.418 |
>
> , where 'half' means we halve all of HLoss's coefficients in our default setting, and 'double' means we double all of the coefficients. We only trained the model for ten epochs, thus the findings may be suboptimal. The approach appears to be robust to hyperparameter adjustment, as the above results are close to each other.
>
> * the concept of deep supervision mentioned here does not appear to be introduced or discussed
>
> We sincerely apologize for the inconvenience.  In our initial publication, we discuss deep supervision in the introduction.
> "We also incorporate intermediate
> supervision and regularization on each layer’s output to ensure the generated heatmaps
> align with human intuition. " However, we removed it in the revised manuscript. Sorry for the inconvenience again.
>
> * there is no guidance on how these hyperparameters could be selected for other tasks,
>
> We concur with the reviewer. Nevertheless, we would also want to stress that we place greater emphasis on ICH subtype segmenation in our updated version. For instance, we include words like  "In this work we are interested
> in ICH subtype segmentation. Hematoma volume and hematoma expansion are essential
> predictors of mortality and outcome (Li et al., 2024). Previous studies (Scherer et al., 2016)
> have shown that the ABC/2 method, ..."
> We believe weakly supervised ICH subtype segmentation is not an easy task. To the best of the authors' understanding, we did not find topics related to weakly supervised ICH subtype segmentation in the literature.
>
> * The visualizations are not presented using a unified color map ,  Guided Backpropagation appears to perform well qualitatively in the visual results
>
> We adhere to the official implementation for integrated gradients. If we utilize a color map for guided backpropagation, all of the photos will be red. The issue with guided backpropagation is that all of the pixel values are too similar, making it difficult to choose the relevant area by only selecting a threshold. In section 5 we wrote "While many of these heatmaps seem reasonable, their performance is surprisingly inferior
> to that of RDNet alone. This can be possibly due to too many noises in the heatmaps." We appreciate the opportunity to clarify it once more.
>
>  * The proposed approach appears similar to that described
>
> To the best of the authors' knowledge, the suggested article had no guidelines about weakly supervised segmentation. We concur with the reviewer, though, that it is feasible to accomplish this with a few adjustments. We would like to emphasize that we are performing weakly supervised segmentation rather than explanation in our work.
>
> Moreover, we believe that these two works have very distinct goals. We are attempting to perform fine-grained segmentation on irregular regions such as SAH and SDH. However, fundus, OCT, and chest X-ray images are the  selected tasks in the recommended paper. Low-resolution latent spaces can be used for these tasks since the shapes of anomalous regions are regular.
>
> Another significant distinction between these two publications is that in our study, we attempt to empirically persuade the readers that fine-grained architectures encode richer information, even when the majority of classifiers obtain similar AUCs.  We did not find any related information in the recommended paper.
>
> Because of the aforementioned factors, we believe that these two papers are best suited for distinct objectives and that a direct comparison is difficult. We appreciate the reviewer allowing us to clarify it once more.
>
> * upsampling may introduce noise.
>
> We apologize but we are not sure. For ICH subtype segmenation, we did not observe this phenomenon.

---

> > ### Author Response · Authors · 2026-01-30
> > **Update on the performance of the model using a single classification layer at the final decoder stage.**
> >
> > Dear Reviewer ucPs,
> >
> > Thank you for your suggestions. Regarding to your question about the performance of the model using a single classification layer at the final decoder stage, we summarize as follows:
> >
> >  |Model | Normal| IPH | IVH  | SAH | SDH | EDH|AVG|
> >   |------|------|------| ------|------|------| ------|------|
> >   | Single Class Layer | 0.991 | 0.057 | 0.148 | 0.099 | 0.065 | 0.043 | 0.234 |
> >
> > We obtained an AUC of 0.961 after training the model for ten epochs. Nevertheless, the results for weakly supervised segmentation are unsatisfactory, as shown in the table above. We think this is because the model did not make full use of the encoder's deep layer representations.

---

### Official Review · Reviewer_q7np · 2026-01-09

**Confidence:** 3
**Preliminary Rating:** 3

**Summary:**

In this work, authors study diverse neural network architectures and compare their Class Activation Mapping (namely, GradCam) for medical image classification. To conduct such an analysis, experiments in intracerebral hemmorage classification in CT images are performed. Based on the observed performance, a Unet-base architecture with higher latent spaces is further proposed towards generating refined class activations.

**Strengths:**

- Authors address a critical need in model interpretability, as the usage of class activation maps.
- A UNet architecture that incorporates LogSumExp functions, and enables higher-dimensional latent spaces is proposed.
- Authors show improvements in classification performance when comparing their model againt diverse resnet and densenet CNNs.

**Weaknesses:**

- There are no ablation studies of the introdiced the model architecture to get better insights on where the gains come from. While authors include a section entitle "ablation study", such an analysis does not exist.
- Authors did not explore diverse versions og GradCam, even though the vast alternative availables
- For concluding results, it would have been relevant to at least include a second dataset for the experiments. This could have shown weather the findings are not task/dataset-specific.
- Dice tables could benefit from any dispersion metric (as standard-deviation) to get inisgihts into results distribution.

**Detailed Comments:**

- The text is not easy to follow. For instance, the introduction contains already details about the results, making it hard to understand.

**Justification Of The Preliminary Rating:**

There are observable weaknesses in the paper that could have been addressed through a better experimental design. However, not being an expert in this field, i leave the opportunity to authors to explain their choices and enhance the work.

**Questions To Address In The Rebuttal:**

- Are shown Dice values mean/median? Please clarify.
- Raw CT images in Figures (e.g. Fig. 1, but also others) have an overlayed map on it, making it hard to understand where the lesions are located. Since the activation maps are already  shown, please show the raw CT images without overlays.
-Given my previous point, it seems hard to understand if the generated "lesions", are indeed lesions. For instance, in Figure 5, 2nd column, the displayed structure seems to be the anatomic circle of willis. Showing the raw CT will benefit its understanding.
- What is the "constrain" show in the network architecture? Please clarify all elements shown on it. The caption is very poor.
- Figures and tables should be self-contained. However, their captions have little information. For instance, what represents * in densenet across all tables?

---

> ### Author Response · Authors · 2026-01-24
> **The manuscript has been changed, with extra information added to the caption.**
>
> Dear Reviewer q7np,
>
> Thanks for your informative suggestions. We summarize our responds as follows:
>
> * Are shown Dice values mean/median? Please clarify.
>
> The results in Table 2 is the Dice values as a whole (viewing the dataset as one object). Now we add the Dice values for the instance level in Table 3.  For data preparation, please refer to the sentences highlighted in yellow in section 4.
>
> * please show the raw CT images without overlays.
>
> We concatenate three consequent slices into one. Sorry about the confusion. It is now corrected.
>
> * it seems hard to understand if the generated "lesions", are indeed lesions. For instance, in Figure 5, 2nd column, the displayed structure seems to be the anatomic circle of willis.
>
> Yes, the high-intensity spots surrounding the circle of Willis are SAH. That is precisely the impetus for our work: subtype segmentation is not difficult for humans. As shown, hyper-intensity patches around the circle of Willis are unquestionably SAH. Is it possible to create subtype segmentation without segmentation labels?
>
> * What is the "constrain" show in the network architecture?
>
> Thank you for reminding us. We will now modify the word "constraint" to "HLoss" as described in section 3.2.
>
>  * what represents * in densenet across all tables?
>
> Thank you for the corrections. * Indicates densenet without using the pretrained weight. As shown in Fig. 6, even though the AUC of densenet and densenet* are identical (in Table 1), densenet's outputs encode more information.
>
> * There are no ablation studies of the introdiced the model architecture to get better insights
>
> Thank you for reminding us. Originally, we just showed the figures in Fig. 5. The dice values are now displayed in the eighth row of Table 2.
>
> * Authors did not explore diverse versions of GradCam,
>
> As proposed by Reviewer ucPs, we have implemented guided backpropagation and integrated gradients as baselines. The results are shown in Table 2 and Figure 5. Surprisingly, the results are poorer than GradCAM.
>
> * This could have shown weather the findings are not task/dataset-specific.
>
> We agree with the reviewer that the method is presently only demonstrated to operate on the ICH dataset, and we need to study the generalizability of our methods on other datasets. Unfortunately, due to the time constraints, we have not completed it yet.
>
> We added sentences in the first paragraph of Section 1 (highlighted in yellow) indicating that our primary objective is ICH subtype segmentation. We will do more investigation on the generalization as our future work.

---

### Official Review · Reviewer_6TZX · 2026-01-10

**Confidence:** 3
**Preliminary Rating:** 3
**Final Rating:** 3

**Summary:**

The paper addresses an interesting current problem: heatmaps for visualization are commonly derived from 16x16 or similar aggregated representations and are "approximate" and interpolated. They propose a different way to produce explainability trough a UNet-like architecture for classification, with deep supervision generating high resolution, sharp, regularized heatmaps.

**Strengths:**

The results clearly demonstrate more detailed and sharper GradCAM-style visualizations compared to standard low-resolution CAMs. The ideas presented in the paper have potential and were clearly thought out with care, with LSE generating activations at multiple levels (deep-supervision-like?) that are detailed and supervised only by classification labels. The math and reasoning in this manuscript appear technically sound to me.

**Weaknesses:**

Writing can be significantly improved in flow and organization, the introduction already talks about results, method description and experiments all at once. Sections are disconnected, which generates confusion for the reader.

The related works Is also confusing, mentioning some segmentation works and gradcam. are there not other works attempting to compete with/improve gradCAM?

What is the objective? Weak segmentation supervision, or improved explainability? This is very unclear, and makes the rest of the paper hard to follow and confusing.

Sections 3.3 is confusing and should be rewritten. Section 6 is unnecessary in its current form, and this information should be elsewhere.

**Detailed Comments:**

In the introduction, " (Zhou et al., 2016; Selvaraju et al., 2017) are currently the most
widely used techniques. " avoid using references as the subject, prefer the name of the techniques when possible.

Tables need lines. Some notations for equations could be introduced in Figure 3.

**Justification Of Final Rating:**

I commend the effort to improving the paper's writing and organization. Unfortunately my final rating is still the same. Would be a reject by the organization, formatting and paper flow aspect (2/5), and is bumped to a borderline (3/5) due to the interesting method proposal.

**Justification Of The Preliminary Rating:**

The paper is not very well written and formatted, information is all over the place frankly. It needs a better Section structuring and flow. In its current state the amount of work necessary for an acceptable version is high.

The idea and proposed architecture is interesting and appears to work well, which raises this from a reject for my initial rating to borderline.

**Questions To Address In The Rebuttal:**

Dice scores for vanilla GradCAM applied to the same pretrained encoders are missing, making it difficult to isolate whether improvements stem from the proposed architecture or simply from higher-resolution feature maps. Have the authors evaluated Dice for this baseline case?

---

> ### Author Response · Authors · 2026-01-24
> **Dice scores can be found in Table 2; manuscripts are reorganized**
>
> Dear Reviewer 6TZX,
>
> Thanks for your informative suggestions. We have significantly updated our manuscripts. The responses to your inquiries are outlined below:
>
> * Dice scores for vanilla GradCAM applied to the same pre-trained encoders are missing
>
> We used RDNet (illustrated in Sec. 3.1) as our encoder, the performances of RDNet can be found in Table 1. , 2. , and 3.
>
> The majority of the baselines presented in Tables 1, 2, and 3 are GradCAMs from various networks. We want to stress that our aim is not to only present 'weaker' baselines. Rather, we intend to demonstrate that, as outlined in the summary, although all models exhibited comparable classification accuracy, the quality of the heatmaps produced differed greatly. For instance, the performance of RDNet and ResNetRS notably surpasses that of traditional models. Our primary contribution lies in the addition of a simple decoder, allowing its weights to be guided by these detailed encoders, resulting in more precise outputs.
>
> * avoid using references as the subject, prefer the name of the techniques when possible.
>
> We sincerely thank the reviewer for the correction; we have already addressed these errors.
>
> * What is the objective? Weak segmentation supervision, or improved explainability?
>
> We appreciate the review's reminder. We've added sentences to the first paragraph (highlighted in yellow) to illustrate our motivation: we're interested in ICH subtype segmentation. Clinically, the ABC/2 approach (which estimates lesion volume by multiplying the biggest total clot diameter by the appropriate perpendicular clot diameter and the number of slices) is used. However, it is extremely erroneous. We focus on weakly supervised segmentation because fully supervised annotations are time-consuming.
>
> * Sections 3.3 is confusing and should be rewritten
>
> We rewrote Section 3.3. Our idea is simple: we discovered that the subtype heatmaps generated by the classifiers are nearly mutually exclusive. As a result, we construct subtype heatmaps and use argmax to make our final prediction. However, we discovered that Grad-CAM creates too many false positives, therefore we designed the class and mask criteria. If the subtype classifier output falls below the class threshold, the heatmap for that subtype will not be generated. The mask threshold indicates that we set heatmaps with pixel values less than this threshold to zero.
>
> * Section 6 is unnecessary in its current form, and this information should be elsewhere.
>
> We reworked sections 6 to focus on weakly supervised ICH segmentation. We withhold the line "while all models showed similar classification accuracy, the quality of the heatmaps they generated varied significantly" since we did not find similar comments in the literature. Are these sentences taking away from the paragraph's main point?
>
> * The related works Is also confusing, mentioning some segmentation works and gradcam
>
> We added phrases explaining our purpose in the opening paragraph (highlighted in yellow) to stress our interest in ICH subtype segmentation; we hope this makes the text more consistent.
>
> * the introduction already talks about results, method description and experiments all at once
>
> We transfer the technically related paragraph to Section 3.1.
>
> * are there not other works attempting to compete with/improve gradCAM?
>
> Section 2.2 now includes some additional content. As recommended by Reviewer ucPs, we provide integrated gradient and guided backpropagation as additional baselines. The findings are given in Table 2 and Figure 5. We discovered that the performances are inferior to those of GradCAM.
>
> We include section 2.3 on weakly supervised segmentation. Even without foundation models, there are many works that are performing effectively on PASCAL VOC2012. As a result, I believe these works may also be applicable in our circumstance. However, because earlier research used CAM predictions as initial pseudo labels, our work is complimentary to them and can be seen as superior starting pseudo labels. We discovered one paper on weakly supervised ICH segmentation (binary, not subtype segmentation) using the Swin transformer, which we now include in section 2.3.
>
> * Tables need lines. Some notations for equations could be introduced in Figure 3.
>
> Lines are added. We updated the word "constraint" in Figure 3 to "HLoss".
>
> We thank the reviewer for informing us of the paper organization. We rewrote our manuscript.

---

### Author Rebuttal · Authors · 2026-01-24

**Rebuttal:**

Dear Reviewers,

We express our gratitude to the reviewer for their valuable feedback regarding the manuscript's structure. In response, we have undertaken revisions, including the incorporation of motivations for ICH subtype volume estimation within the introduction. Furthermore, the technical discussion previously situated in the introduction has been relocated to Section 3.1. We have also introduced two additional baselines, namely guided backpropagation and integrated gradients, and have included Section 2.3, which addresses weakly supervised segmentation. It is our sincere hope that this revised version exhibits improved organization and enhanced readability for our intended audience.

The majority of the baselines presented in Tables 1, 2, and 3 are GradCAMs from various networks. Despite comparable classification performance, there are substantial differences in their  representation powers. Our main contribution is to harness such representation by adding a decoder with simple structure such that their weights are guided by the fine-grained encoder when retraining.

In this study, our primary emphasis is on the ICH dataset. We concur with the reviewers that validating our approach on an additional dataset is essential for ensuring generalizability. However, due to time constraints, we have not completed this yet. We will continue to explore generalizability in our future research.

We genuinely appreciate all the feedback provided by the reviewers once more.

**Supporting Material:**

/attachment/69bf945ee9ef0ec9850bc78ec19244bdbd8dd3e3.pdf

---

### Meta-Review · Area_Chair_ghUB · 2026-02-09

**Recommendation:** Reject
**Confidence:** 4

**Metareview:**

This paper proposes an interesting architecture for weakly supervised localization, with reviewers agreeing on the promise of the core idea. However, across reviews, substantial concerns remain regarding clarity, organization, methodological justification (architecture and hyperparameters), limited evaluation scope (single dataset), and insufficient ablations, leaving the contribution not yet mature enough for acceptance.

---

### Decision · Program_Chairs · 2026-02-14

Accept (Poster)